# The relationship between social safeness and pleasure and resilience levels among university athletes: *A descriptive study*

**Veysel Temel**[1]*, Nurhan Hümeyra Özçelik[2]

**1** Faculty of Sports Sciences, Karamanoğlu Mehmetbey University, Karaman, Türkiye, **2** Faculty of Health Sciences, Karamanoglu Mehmetbey University, Karaman, Türkiye

* veyseltemel@kmu.edu.tr

## Abstract

This study investigates the correlation between social safeness and pleasure and resilience among university students engaged in sports. A total of 350 participants (mean age, 21.09 ± 3.12 years), comprising 239 females and 111 males, were included in the sample. Data collection utilized the "Social Safeness and Pleasure and Resilience Scales". Normality of data was assessed using skewness and kurtosis tests, confirming normal distribution. Pairwise comparisons were conducted using t-tests, multiple comparisons via one-way ANOVA, and relationship analysis employed Pearson Product Moment Correlation. Findings indicate participants scored above mid-level on the social safeness and pleasure scale (Mean = 2.6), and resilience scale (Mean = 3.7). Specifically, social support dimension mean score was above mid-level (Mean = 4.01), emotional coping (Mean = 3.4), and situational coping (Mean = 3.8). Pearson Product Moment Correlation revealed a statistically significant positive relationship between social safeness and pleasure scale scores and resilience scale scores (r = .652, p < .001). This descriptive study sheds light on the interplay between social and emotional factors and resilience among university athletes.

## Introduction

The word "sport" finds its origins in Latin, where it originally meant "to linger, to separate, and to move away," evolving over time into the concept we know today [1]. Beyond its linguistic roots, sports encompass activities that go far beyond physical exercise. They are deliberate pursuits aimed at enhancing both physical and mental health, fostering personality development, and equipping individuals with essential skills and knowledge for navigating their environment [2]. Engaging in sports offers a multifaceted approach to personal growth and societal integration. It promotes positive outlooks, effective stress management, social harmony, and acceptance within communities [3]. Physical activity refers to any bodily movement produced by skeletal muscles that requires energy expenditure. This includes a wide range of activities such as walking, cleaning, and even incidental movements during daily tasks [4]. Physical activity can be of low, moderate, or high intensity and does not necessarily have a structured

**Data Availability Statement:** The data is included in the publication. It can also be sent via spss when requested. The data has been organized and

anonymized to include all raw data and analysis results obtained as a result of the study. Contact Information For the Data Access Committee, Ethics Committee; 1- Prof. Dr. Murat TEKİN, Karamanoglu Mehmetbey University, Faculty of Sports Sciences, Department Head And Ethics Committee Member, murattekin76@gmail.com, +90.338.2262117 2- Prof. Dr. Elif BİROL, Karamanoglu Mehmetbey University, Faculty of Sports Sciences, Department Head And Ethics Committee Member, elifaydin@kmu.edu.tr, +90.338.2264217 3- Prof. Dr. Sefa LÖK, Karamanoglu Mehmetbey University, Faculty of Sports Sciences, Department Head And Ethics Committee Member, sefalok@gmail.com, +90.338.2262110

**Funding:** The author(s) received no specific funding for this work.

**Competing interests:** The authors have declared that no competing interests exist.

or planned purpose beyond increasing energy expenditure. In contrast, physical exercise is a form of physical activity that is planned, structured, and aimed specifically at improving or maintaining physical fitness. Exercise is characterized by its repetitive nature and is generally performed with specific goals, such as improving cardiovascular health or muscular strength [5]. Examples of physical exercises include activities like weightlifting, running, and swimming. Sport, however, is distinct in that it generally involves organized, competitive activities governed by established rules and regulations. Sports are structured, often include a formal competition element, and are typically played at various levels ranging from recreational to professional [6]. While sport incorporates physical activity and exercise, it also requires a high level of skill, strategy, and often teamwork. For instance, football, tennis, and athletics all fall under the umbrella of sport, but they involve not only physical exertion but also complex tactical elements and competition. Thus, while physical activity and physical exercise are integral for health and well-being, sport specifically refers to structured and competitive endeavors, often underpinned by formalized rules, which distinguish it from general physical activity and exercise. The practice of sports not only enhances individuals' physical well-being but also nurtures their psychological resilience and social confidence. It serves as a catalyst for personal improvement, fostering a sense of self-assurance and empowerment [7]. In understanding the role of sports in society, particularly in fostering social trust and satisfaction, we recognize its profound impact on personal development. Sports provide individuals with opportunities to build trust through shared experiences, collaborative efforts, and mutual respect within their communities [8]. This trust forms the foundation for healthy interpersonal relationships and contributes to a sense of belonging and security among individuals [9]. Moreover, sports serve as a common ground where individuals from diverse backgrounds come together to pursue common goals. This collective effort not only enhances individual well-being but also strengthens community bonds and resilience against challenges [10]. In essence, the practice of sports transcends mere physical activity. It cultivates social cohesion, personal growth, and resilience, making significant contributions to individuals' overall quality of life and societal harmony.

Resilience encompasses various qualities such as psychological strength, endurance, and the ability to rebound from adversity with elasticity [11]. It is not merely an inherent trait but also a developmental process shaped by familial and environmental support systems that aid individuals in surpassing previous levels of functioning after facing crises [12]. In addition to physical recuperation, mental recovery plays a crucial role in athletes' ability to maintain motivation and sustain performance by alleviating stress and mental fatigue [13]. From a psychosocial perspective, engagement in sports facilitates individuals' recovery from daily life challenges, enhancing their sense of social security and overall satisfaction [14]. Social trust, stemming from interpersonal interactions, reflects individuals' confidence in others and significantly influences their cognitive processes and problem-solving abilities [15]. Sports contribute to individuals' sense of social safeness and pleasure, fostering self-confidence and optimism about their ability to shape their futures [16]. This sense of security and positive affect derived from social safeness enables individuals to recover effectively, drawing strength from societal support and mitigating feelings of isolation during challenging times [9]. Research on resilience highlights that individuals who recover swiftly from difficulties possess strong social skills, autonomy, a sense of belonging, and optimism about future prospects [17]. Therefore, sports serve as a platform for developing these attributes through healthy interpersonal relationships and a supportive community environment. In conclusion, integrating sports into daily life not only enhances physical health but also fosters psychological resilience, promotes social trust, and strengthens individuals' ability to overcome adversity. This comprehensive approach underscores the profound impact of sports on personal development and overall well-being. The findings of various studies confirm the Social Safeness and Pleasure

Scale's reliability and validity, highlighting its significant association with psychological health and well-being [7,18]. While resilience has garnered considerable attention in Western societies, research on this topic remains limited in our country [19–21]. Hence, this study utilized both the Social Safeness and Pleasure Scale and the Resilience Scale.

The primary aim of this study was to explore the relationship between individuals' levels of social safeness and pleasure and their resilience. The correlation analysis, such as the Pearson Product Moment Correlation Analysis, is used to assess the strength and direction of relationships between two or more variables [22]. However, it is important to note that correlations alone do not establish causality or direct intervention strategies. By understanding this connection, we gain insights into how social safeness and pleasure influence individuals' ability to navigate and cope with challenging and negative situations. Also, Gender differences have been an important aspect in various fields of research, particularly in studies examining psychological and physiological variables. In sports and physical activity contexts, gender can influence multiple factors such as resilience, social safeness, and emotional coping mechanisms [23,24]. Research has shown that men and women may experience and respond to sports-related stressors differently, which can impact their psychological well-being and coping strategies [25]. Therefore, it is crucial to include gender as a variable when exploring psychological constructs in athletic populations to understand how these differences may affect sports performance and overall well-being [26]. This research contributes to identifying interventions that can enhance and elevate individuals' levels of social safeness and pleasure.

## Materials and methods

### Research design

This study employed relational and descriptive survey methods, widely recognized in the social sciences for their effectiveness in exploring relationships and assessing current conditions. The relational screening method, as described by Karasar, investigates the co-variation between variables, aiming to uncover associations among them. On the other hand, descriptive survey research, following Büyüköztürk et al., provides a snapshot of the current state of affairs within the studied population [27,28].

### Participants

The research group consisted of 350 students from Karamanoğlu Mehmetbey University, distributed across different faculties and sporting habits as follows: Faculty of Sports Sciences (n=87, 24.9%), Faculty of Health Sciences (n=122, 34.9%), and Faculty of Islamic Sciences (n=141, 40.3%). Among them, 200 students (57.1%) were engaged in sports, while 150 (42.9%) were not. The gender distribution was 247 female participants and 103 male participants. The participants' ages ranged from 18 to 50 years, with an average age of 30.28 ± 8.99 years. In this study, participants were categorized based on their frequency of sports participation. The participants were classified as follows: "Seldom" (35 individuals, 10.0%), "Sometimes" (56 individuals, 16.0%), "Usually" (51 individuals, 14.6%), "Often" (38 individuals, 10.9%), and "Always" (20 individuals, 5.7%). The participants were active individuals who engage in sports at sports clubs or sports centers.

According to Table 1, 70.6% of the participants are female, while 29.4% are male, with the majority falling within the 22-24 age range (64.0%). The highest participation rates are observed among students from the Faculty of Islamic Sciences (40.3%) and the Faculty of Health Sciences (34.9%). Among academic departments, Islamic Sciences (40.3%) and Nursing (28.3%) have the largest representation. A total of 57.1% of participants engage in sports, predominantly individual sports (40.9%). Regarding the frequency of sports participation, the

**Table 1. Descriptive statistics of the participants.**

| | | N | % |
|---|---|---|---|
| **Gender** | Female | 247 | 70,6 |
| | Male | 103 | 29,4 |
| **Age** | 18 Yaş - 21 Yaş | 78 | 22,3 |
| | 22 Yaş - 24 Yaş | 224 | 64,0 |
| | 25 Yaş - 27 Yaş | 27 | 7,7 |
| | 28 Yaş + | 21 | 6,0 |
| **Faculty** | Faculty of Sports Sciences | 87 | 24,9 |
| | Faculty of Health Sciences | 122 | 34,9 |
| | Faculty of Islamic Sciences | 141 | 40,3 |
| **Department** | Physical Education and Sports Teaching | 60 | 17,1 |
| | Sports Management | 8 | 2,3 |
| | Coaching | 19 | 5,4 |
| | Nursing | 99 | 28,3 |
| | Nutrition And Dietetics | 13 | 3,7 |
| | Child Development | 10 | 2,9 |
| | Islamic Sciences | 141 | 40,3 |
| **Doing Sports** | Yes | 200 | 57,1 |
| | No | 150 | 42,9 |
| **Sports Type** | Individual | 143 | 40,9 |
| | Team | 57 | 16,3 |
| **Sports Doing Frequency** | Seldom | 35 | 10,0 |
| | Sometimes | 56 | 16,0 |
| | Usually | 51 | 14,6 |
| | Often | 38 | 10,9 |
| | Always | 20 | 5,7 |
| **Do you think it is right to get support from a psychologist when the need arises?** | Yes | 299 | 85,4 |
| | No | 51 | 14,6 |
| **Do you have difficulty in using your leisure time?** | Always | 26 | 7,4 |
| | Sometimes | 240 | 68,6 |
| | Never | 84 | 24,0 |

highest proportion (16.0%) reported engaging in sports "sometimes." Additionally, 85.4% of participants consider seeking psychological support appropriate when necessary, while 68.6% occasionally experience difficulty in utilizing their leisure time.

## Data collection tool

The research utilized two main data collection tools. Firstly, the Social Safeness and Pleasure Scale, adapted into Turkish by Akın et al. (2013), was employed [29]. This scale consists of 11 items designed to measure a single dimension using a 5-point Likert scale. Confirmatory factor analysis (CFA) confirmed the validity of the Turkish version, showing good fit indices (RFI=.95, GFI=.96, SRMR=.042), with factor loadings ranging from .34 to .61. Convergent validity was established through correlations, demonstrating a negative relationship with self-alienation (r=-.18) and a positive correlation with authentic life (r=.35). Internal consistency reliability (Cronbach's α) for the study was high at .90. Secondly, the Resilience Scale, developed by Johnson et al. (2010) [30] and adapted into Turkish by Sarıçam et al. (2012), was employed [8]. This scale comprises 12 items across three sub-dimensions, utilizing a 5-point

Likert scale from (1) I completely disagree to (5) I completely agree. CFA confirmed the fit of the 3-dimensional model with satisfactory indices ($\chi^2$=117.28, df=47, p=0.00000, RMSEA=.060, CFI=.97, NFI=.95, RFI=.95, IFI=.97, GFI=.96, SRMR=.049). Item-total correlations ranged from .38 to .57. Internal consistency reliability coefficients were .91 for total resilience, .77 for emotional coping, and .87 for situational coping. These tools were integral in assessing social safeness, pleasure, and resilience among the participants, offering robust insights into these psychological constructs within the university setting.

**Data collection process.** Prior to commencing the research implementation, necessary approvals were obtained from relevant institutions and individuals. The study proceeded with face-to-face administration of three-part inventories on a voluntary basis among participants selected from the sample group. Detailed explanations were provided regarding the research objectives, content, and instructions for carefully reading and completing the questionnaire.

**Ethical aspects of the research.** The data of the study were collected between May and June 2024 with the ethics committee approval permission from Karamanoğlu Mehmetbey University Social and Human Scientific Research Ethics Committee (Date: 22.04.2024; Decision No: 06-2024/131). Informed written consent was obtained from all participants. Ethical rules were followed during the research. Permission to use the "*Social Safeness And Pleasure Scale*" developed by Gilbert et al [7] and adapted into Turkish by Akın et al. [29] and the "*Resilience Scale*" developed by Gail Wagnild et al. and adapted into Turkish by Terzi was obtained from the first author via e-mail [31]. The authors confirm their specific contributions to the work presented. The authors are in agreement on the conclusions, implications, or opinions stated in the manuscript reported.

## Results

This section presents the findings derived from the analysis of the collected data concerning the variables and hypotheses investigated in the study.

In the Table 2, the normality tests for the Social Safeness and Pleasure scale, as well as the Resilience scale and its sub-dimensions (Social Support, Emotional Coping, Situational Coping), yielded results indicating compliance with normal distribution assumptions (Skewness and kurtosis between -1.5 and +1.5). Additionally, Levene's test results (>.05) confirmed homogeneity of variance-covariance matrices among dependent variables. With all assumptions met, the analysis proceeded. Examining participants' mean scores:

- The average score for Social Safeness and Pleasure was 2.68, indicating a level above the midpoint.

- The Resilience scale's total mean score was 3.7, indicating an above-average level.

- Social Support scored an average of 4.01, Emotional Coping 3.47, and Situational Coping 3.8, all indicating scores above the midpoint.

According to the t-test results presented in Table 3, significant differences were observed in resilience, emotional coping, and situational coping scales based on gender. In terms of resilience, males (mean = 3.8881) showed statistically significantly higher scores compared to females (mean = 3.7050) with a t-value of -2.283 (p < 0.05). Similarly, males scored higher than females in emotional coping (t = -3.026, p = 0.003) and situational coping (t = -2.866, p = 0.003). These findings underscore the impact of gender on psychological resilience and coping strategies, highlighting that males generally exhibit higher levels of resilience and coping skills than females.

**Table 2. Descriptive statistics of social safeness and pleasure and recovery scale.**

|  | N | $\overline{X}$ | Ss | Skewness | Kurtosis | Min. | Max. |
|---|---|---|---|---|---|---|---|
| Social Safeness And Pleasure | 350 | 2,68 | ,79 | -0,73 | 0,73 | ,00 | 4,00 |
| Resilience Total | 350 | 3,76 | ,70 | -0,52 | 0,67 | 1,00 | 5,00 |
| Social Support | 350 | 4,01 | ,77 | -0,87 | 0,77 | 1,00 | 5,00 |
| Emotional Coping | 350 | 3,47 | ,92 | -0,38 | -0,13 | 1,00 | 5,00 |
| Situational Coping | 350 | 3,80 | ,77 | -0,60 | 0,75 | 1,00 | 5,00 |

Table 4 presents the results of one-way ANOVA tests conducted to examine differences in Social Safeness And Pleasure, Resilience Total, Social Support, Emotional Coping, and Situational Coping across different age groups. Here's a concise summary of the findings: Statistically significant differences were found across age groups in Social Safeness And Pleasure, Resilience Total, Social Support, Emotional Coping, and Situational Coping scales ($p < 0.05$). Specifically, younger age groups tended to score lower on these psychological scales compared to older age groups. This trend suggests that as age increases, individuals may exhibit higher levels of social safeness, resilience, social support, and effective coping strategies. These results underscore the potential impact of age on psychological resilience and coping mechanisms.

Table 5 presents findings from one-way ANOVA tests examining differences in Social Safeness And Pleasure, Resilience (total and sub-dimensions), and Social Support across three faculties: Faculty of Sports Sciences, Faculty of Health Sciences, and Faculty of Islamic Sciences. No significant differences were found in Social Safeness And Pleasure perceptions among faculties ($F = 3.73$, $p = 0.051$). However, Resilience Total scores differed significantly ($F = 5.37$, $p = 0.004$), with lower scores observed in the Faculty of Sports Sciences compared to Health Sciences and Islamic Sciences. Social Support showed no significant differences ($F = 2.08$, $p = 0.177$) across faculties. Emotional Coping ($F = 11.25$, $p = 0.001$) and Situational Coping ($F = 5.79$, $p = 0.008$) varied significantly, indicating distinct coping strategies among faculties. In summary, while perceptions of social safeness and support were consistent, resilience levels and coping strategies varied across academic disciplines. Tailored interventions could enhance these skills, especially in faculties showing lower resilience and specific coping challenges.

**Table 3. t-Test results conducted to determine whether participants' social safeness and pleasure scale and resilience scale total and sub-dimension scores differ according to gender variable.**

|  | Groups | N | $\overline{X}$ | Ss | Shg | T Test | | |
|---|---|---|---|---|---|---|---|---|
|  |  |  |  |  |  | T | Sd | P |
| Social Safeness And Pleasure | Female | 247 | 2,67 | ,78 | ,050 | -,399 | 348 | ,690 |
|  | Male | 103 | 2,70 | ,81 | ,077 |  |  |  |
| Resilience Total | Female | 247 | 3,70 | ,70 | ,045 | -2,283 | 348 | **,023\*** |
|  | Male | 103 | 3,89 | ,69 | ,065 |  |  |  |
| Social Support | Female | 247 | 4,02 | ,75 | ,048 | ,213 | 348 | ,831 |
|  | Male | 103 | 3,99 | ,83 | ,079 |  |  |  |
| Emotional Coping | Female | 247 | 3,37 | ,91 | ,059 | -3,026 | 348 | **,003\*** |
|  | Male | 103 | 3,69 | ,90 | ,086 |  |  |  |
| Situational Coping | Female | 247 | 3,72 | ,79 | ,051 | -2,866 | 348 | **,003\*** |
|  | Male | 103 | 3,98 | ,72 | ,068 |  |  |  |

\* $p < 0.05$.

**Table 4. Results of One-Way Analysis of Variance (One-Way ANOVA) conducted to determine whether participants' social safeness and pleasure scale and resilience scale total and sub-dimension scores differ according to age variable.**

| | *f, x ve ss Values* | | | | One-Way ANOVA Result | | | | | | |
|---|---|---|---|---|---|---|---|---|---|---|---|
| | Group | N | $\overline{X}$ | Ss | V. Set | KT | Sd | KO | F | P | Dif. |
| **Social Safeness And Pleasure** | 01 | 78 | 2,57 | ,83 | ,09 | 7,68 | 3 | 2,56 | 4,18 | **,006*** | **3-1/2** |
| | 02 | 224 | 2,65 | ,78 | ,05 | 211,73 | 346 | ,612 | | | |
| | 03 | 27 | 3,08 | ,64 | ,10 | 219,41 | 349 | | | | |
| | 04 | 21 | 2,69 | ,84 | ,21 | | | | | | |
| **Resilience Total** | 01 | 78 | 3,78 | ,65 | ,07 | 10,30 | 3 | 3,43 | 7,34 | **,001*** | **3-1/2** |
| | 02 | 224 | 3,66 | ,72 | ,05 | 161,95 | 346 | ,468 | | | |
| | 03 | 27 | 4,16 | ,56 | ,09 | 172,25 | 349 | | | | |
| | 04 | 21 | 4,08 | ,57 | ,14 | | | | | | |
| **Social Support** | 01 | 78 | 4,03 | ,73 | ,08 | 11,56 | 3 | 3,85 | 6,75 | **,001*** | **3-1/2** |
| | 02 | 224 | 3,90 | ,80 | ,05 | 197,59 | 346 | ,571 | | | |
| | 03 | 27 | 4,46 | ,55 | ,09 | 209,15 | 349 | | | | |
| | 04 | 21 | 4,23 | ,64 | ,16 | | | | | | |
| **Emotional Coping** | 01 | 78 | 3,53 | ,89 | ,10 | 11,02 | 3 | 3,67 | 4,48 | **,004*** | **3-2** |
| | 02 | 224 | 3,35 | ,94 | ,06 | 283,67 | 346 | ,820 | | | |
| | 03 | 27 | 3,86 | ,81 | ,13 | 294,69 | 349 | | | | |
| | 04 | 21 | 3,81 | ,73 | ,18 | | | | | | |
| **Situational Coping** | 01 | 78 | 3,78 | ,69 | ,08 | 9,15 | 3 | 3,05 | 5,26 | **,001*** | **3-2** |
| | 02 | 224 | 3,72 | ,81 | ,05 | 200,60 | 346 | ,580 | | | |
| | 03 | 27 | 4,16 | ,66 | ,10 | 209,75 | 349 | | | | |
| | 04 | 21 | 4,20 | ,60 | ,15 | | | | | | |

* p< 0.05 **Groups;** 01=18-21 Age, 02=22-24 Age, 03=25-27 Age and 04=28 Age +.

**Table 5. Results of One-Way Analysis of Variance (One-Way ANOVA) conducted to determine whether participants' social safeness and pleasure scale and total and sub-dimension scores of the resilience scale differ according to faculty variable.**

| | *f, x ve ss Values* | | | | One-Way ANOVA Result | | | | | | |
|---|---|---|---|---|---|---|---|---|---|---|---|
| | Group | N | $\overline{X}$ | Ss | V. Set | KT | Sd | KO | F | P | Dif. |
| **Social Safeness And Pleasure** | 1 | 87 | 2,76 | ,82 | ,08 | 3,73 | 2 | 1,868 | 3,005 | ,051 | - |
| | 2 | 122 | 2,53 | ,82 | ,08 | 215,68 | 347 | ,622 | | | |
| | 3 | 141 | 2,74 | ,74 | ,06 | 219,41 | 349 | | | | |
| **Resilience Total** | 1 | 87 | 3,96 | ,70 | ,07 | 5,37 | 2 | 2,685 | 5,582 | ,004* | 1-2/3 |
| | 2 | 122 | 3,64 | ,73 | ,07 | 166,88 | 347 | ,481 | | | |
| | 3 | 141 | 3,73 | ,66 | ,05 | 172,25 | 349 | | | | |
| **Social Support** | 1 | 87 | 4,10 | ,84 | ,09 | 2,08 | 2 | 1,039 | 1,741 | ,177 | |
| | 2 | 122 | 3,90 | ,79 | ,07 | 207,07 | 347 | ,597 | | | |
| | 3 | 141 | 4,03 | ,71 | ,06 | 209,15 | 349 | | | | - |
| **Emotional Coping** | 1 | 87 | 3,75 | ,85 | ,09 | 11,25 | 2 | 5,627 | 6,889 | ,001* | 1-2/3 |
| | 2 | 122 | 3,29 | ,96 | ,09 | 283,44 | 347 | ,817 | | | |
| | 3 | 141 | 3,44 | ,89 | ,07 | 294,69 | 349 | | | | |
| **Situational Coping** | 1 | 87 | 4,02 | ,72 | ,07 | 5,79 | 2 | 2,897 | 4,929 | ,008* | 1-2/3 |
| | 2 | 122 | 3,72 | ,83 | ,08 | 203,96 | 347 | ,588 | | | |
| | 3 | 141 | 3,73 | ,74 | ,06 | 209,75 | 349 | | | | |

* p< 0.05 **Goups:** 1=Faculty of Sports Sciences, 2=Faculty of Health Sciences, 3=Faculty of Islamic Sciences.

Table 6 presents the results of one-way ANOVA tests examining differences in Social Safeness And Pleasure, Resilience (total and sub-dimensions), and Social Support across different academic departments: Physical Education and Sports Teaching, Sports Management, Coaching, Nursing, Nutrition And Dietetics, Child Development, and Islamic Sciences. Significant differences were found in Social Safeness And Pleasure (F = 12.417, p = 0.003), Resilience Total (F = 13.649, p < 0.001), Social Support (F = 13.631, p = 0.001), Emotional Coping (F = 22.119, p < 0.001), and Situational Coping (F = 12.175, p = 0.002) scores among the

**Table 6. Results of One-Way Analysis of Variance (One-Way Anova) conducted to determine whether participants' social safeness and pleasure scale and total and sub-dimension scores of the resilience scale differ according to the department variable.**

| | *f, x ve ss Values* | | | | One-Way ANOVA Result | | | | | | |
|---|---|---|---|---|---|---|---|---|---|---|---|
| | Group | N | $\overline{X}$ | Ss | V. Set | KT | Sd | KO | F | P | Dif. |
| **Social Safeness And Pleasure** | 1 | 60 | 2,76 | ,83 | ,11 | 12,417 | 6 | 2,070 | 3,429 | ,003* | 6-3/4 |
| | 2 | 8 | 2,74 | ,74 | ,06 | 206,998 | 343 | ,603 | | | |
| | 3 | 19 | 2,51 | ,80 | ,08 | 219,415 | 349 | | | | |
| | 4 | 99 | 2,33 | ,89 | ,25 | | | | | | |
| | 5 | 13 | 3,17 | ,72 | ,27 | | | | | | |
| | 6 | 10 | 3,13 | ,64 | ,15 | | | | | | |
| | 7 | 141 | 2,14 | ,78 | ,29 | | | | | | |
| **Resilience Total** | 1 | 60 | 4,02 | ,65 | ,08 | 13,649 | 6 | 2,275 | 4,920 | ,001* | 1-3 |
| | 2 | 8 | 3,73 | ,65 | ,05 | 158,603 | 343 | ,462 | | | 6-3/4 |
| | 3 | 19 | 3,63 | ,71 | ,07 | 172,252 | 349 | | | | |
| | 4 | 99 | 3,43 | ,71 | ,19 | | | | | | |
| | 5 | 13 | 4,21 | ,75 | ,28 | | | | | | |
| | 6 | 10 | 4,20 | ,79 | ,19 | | | | | | |
| | 7 | 141 | 3,31 | ,67 | ,25 | | | | | | |
| **Social Support** | 1 | 60 | 4,22 | ,74 | ,10 | 13,631 | 6 | 2,272 | 3,985 | ,001* | 1-4 |
| | 2 | 8 | 4,03 | ,71 | ,06 | 195,517 | 343 | ,570 | | | 4-2/6 |
| | 3 | 19 | 3,91 | ,79 | ,08 | 209,147 | 349 | | | | |
| | 4 | 99 | 3,29 | ,91 | ,25 | | | | | | |
| | 5 | 13 | 4,28 | ,73 | ,27 | | | | | | |
| | 6 | 10 | 4,32 | ,82 | ,20 | | | | | | |
| | 7 | 141 | 3,61 | ,77 | ,29 | | | | | | |
| **Emotional Coping** | 1 | 60 | 3,82 | ,74 | ,10 | 22,119 | 6 | 3,687 | 4,639 | ,001* | 1-3/7 |
| | 2 | 8 | 3,43 | ,89 | ,07 | 272,574 | 343 | ,795 | | | 6-3/7 |
| | 3 | 19 | 3,27 | ,93 | ,09 | 294,693 | 349 | | | | |
| | 4 | 99 | 3,33 | ,91 | ,25 | | | | | | |
| | 5 | 13 | 3,93 | 1,06 | ,40 | | | | | | |
| | 6 | 10 | 4,03 | ,97 | ,24 | | | | | | |
| | 7 | 141 | 2,71 | 1,04 | ,39 | | | | | | |
| **Situational Coping** | 1 | 60 | 4,04 | ,72 | ,09 | 12,175 | 6 | 2,029 | 3,523 | ,002* | 1-4 |
| | 2 | 8 | 3,72 | ,74 | ,06 | 197,579 | 343 | ,576 | | | 6-3/4 |
| | 3 | 19 | 3,70 | ,83 | ,08 | 209,754 | 349 | | | | |
| | 4 | 99 | 3,67 | ,72 | ,20 | | | | | | |
| | 5 | 13 | 4,43 | ,73 | ,28 | | | | | | |
| | 6 | 10 | 4,26 | ,67 | ,16 | | | | | | |
| | 7 | 141 | 3,61 | ,61 | ,23 | | | | | | |

* p< 0.05, Goups: 1=Physical Education and Sports Teaching, 2=Sports Management, 3=Coaching, 4=Nursing.

5=Nutrition And Dietetics, 6=Child Development, 7=Islamic Sciences.

departments. These results indicate that perceptions of social safeness, resilience levels, and coping strategies vary significantly based on the academic discipline studied. In summary, the findings highlight the need for tailored support strategies that address the specific psychological needs of students across different departments within the academic setting.

Table 7 summarizes the findings of t-tests assessing differences in Social Safeness And Pleasure, Resilience (total and sub-dimensions), and Social Support between participants who engage in sports and those who do not. There were no significant differences in Social Safeness And Pleasure (t = 1.215, p = 0.225) and Social Support (t = 0.843, p = 0.400) based on sports participation. However, significant differences were found in Resilience Total scores (t = 4.181, p < 0.001), Emotional Coping (t = 4.694, p < 0.001), and Situational Coping (t = 5.030, p < 0.001) between sports participants and non-participants. Participants involved in sports showed higher resilience levels and better coping strategies compared to those who did not engage in sports. In conclusion, while sports participation did not significantly impact perceptions of social safeness and support, it was associated with higher resilience and more effective coping skills, suggesting the beneficial role of sports in enhancing these psychological attributes.

When the table is examined, as a result of the Pearson Product Moment Correlation analysis conducted to determine the relationship between the scores obtained from the social safeness and pleasure scale dimension and the resilience scale test scores, it was determined that there was a statistically significant positive relationship between the scores at the p <.01 level (r = . 515; p>.05).

## Discussion

This section interprets and discusses the findings obtained according to the research results. The findings obtained were supported by the findings related to social safeness pleasure and Resilience in the relevant literature. In the literature reviewed, no study evaluated the research on social safeness pleasure and resilience together. From this perspective, it is thought that it can greatly contribute to the field. In addition, independent studies on the subjects of social safeness pleasure and resilience were also found, and the current study was tried to be supported with the findings of these studies. The current study in the Table 2 reveals that the social safeness and pleasure level average scores of students studying at the faculty of sports sciences,

**Table 7. T-Test results conducted to determine whether the total and sub-dimension scores of the participants in the feeling of social safeness and pleasure scale and the resilience scale differ according to the variable of doing sports.**

|  | Group | N | $\overline{X}$ | Ss | Shg | T Test | | |
|---|---|---|---|---|---|---|---|---|
|  |  |  |  |  |  | T | Sd | p |
| Social Safeness And Pleasure | Yes | 200 | 2,73 | ,80 | ,06038 | 1,215 | 348 | ,225 |
|  | No | 150 | 2,62 | ,78 | ,05938 |  |  |  |
| Resilience Total | Yes | 200 | 3,91 | ,66 | ,05035 | 4,181 | 348 | ,001* |
|  | No | 150 | 3,60 | ,70 | ,05346 |  |  |  |
| Social Support | Yes | 200 | 4,04 | ,79 | ,05970 | ,843 | 348 | ,400 |
|  | No | 150 | 3,97 | ,75 | ,05730 |  |  |  |
| Emotional Coping | Yes | 200 | 3,69 | ,85 | ,06454 | 4,694 | 348 | ,001* |
|  | No | 150 | 3,24 | ,92 | ,07038 |  |  |  |
| Situational Coping | Yes | 200 | 4,00 | ,73 | ,05495 | 5,030 | 348 | ,001* |
|  | No | 150 | 3,60 | ,76 | ,05840 |  |  |  |

p< 0.05.

health sciences, and faculty of Islamic sciences are above the mid-level. It is thought that the reason for this is essentially sports. Individuals who do sports become involved in new environments and share the same social environment, thus increasing their social relationships. This has a positive impact on people's self-confidence levels. As a result of the literature review, Akın et al. argue that sports can be a social activity in which there is defeat and renewal rather than an event that happens on its own [29]. We can understand that humans should be considered social beings and should interact with other people to meet their physiological and psychological needs. According to Aslan, social ties form the basis of human relations [32]. According to Akbaş, one of the important factors affecting human relations is the concept of social trust [33]. Accordingly, we can say that individuals who do sports feel socially safer due to strengthening their relationship with society [34]. Özsarı et al., according to their study on athletes interested in combat sports, concluded that the participants' social safeness and pleasure levels were above the mid-level [35]. Such findings are consistent with literature supporting the positive effects of sports on individuals' social and emotional health. It has been stated in various studies that sports improve individuals' social skills, expand their social networks and increase their psychological resilience [36,37].

The contribution of the sense of trust to social integrity should not be ignored. As Gibbons states, trust strengthens relationships between individuals and is important for every layer of social life [10]. According to Sarıçam's statement, this feeling is a concept whose foundations are laid in infancy and shaped throughout life [8]. It also affects business and academic life. It is thought that individuals who do sports will encounter positive results in this direction. As explained by Gilbert et al., social trust is a result experienced by individuals living in societies where peace, trust and warm relationships are established [9]. It is about individuals belonging and being accepted as part of society. This feeling encourages people to trust each other, cooperate and contribute positively to society. Based on this, it is thought that an environment of peace and security can be achieved by encouraging societies to do sports.

The results of this study reveal a statistically significant positive difference in terms of resilience between male and female participants, as indicated in Table 3. Additionally, men exhibited significantly higher scores in the Emotional Coping and Situational Coping sub-dimensions of resilience. This finding suggests that gender may play a crucial role in determining how individuals cope with stress and adversity. Several factors may explain these gender differences in resilience. From a biological perspective, men and women exhibit differences in genetic makeup and hormone levels, which can influence stress responses and coping mechanisms [38]. Men, for instance, may have higher levels of testosterone, which has been linked to increased resilience and a tendency to face stressors with a more assertive approach [39]. Additionally, societal norms and expectations often dictate that men should exhibit stoicism and refrain from openly expressing emotional vulnerabilities. This may lead men to develop different coping strategies, such as emotional suppression, which could enhance their resilience in certain contexts [40]. The societal roles assigned to different genders also shape how men and women approach stress and coping. In societies where traditional gender roles are emphasized, men may be more likely to engage in physical activities, including sports, which have been shown to improve emotional regulation and coping abilities [36]. Physical activity is known to foster resilience by promoting psychological well-being through increased self-esteem, social support, and the release of endorphins [41]. Consequently, men's greater involvement in physical activities, along with societal expectations to maintain a physically active and healthy lifestyle, may contribute to their higher resilience scores. Supporting this, research by Gençoğlu et al. on physical education students at Erzurum Technical University found that gender significantly affected resilience levels, and factors such as physical activity and lifestyle choices were also influential [42]. However, the study by Reed et al. did not find a significant difference

between men and women in terms of general resilience scores, which suggests that the role of gender in resilience may vary across different populations and contexts [43]. Moreover, while the gender differences in emotional and situational coping observed in this study are noteworthy, it is essential to consider the complexity of these relationships. Further research is needed to explore how gender interacts with other variables, such as personality traits, socio-economic status, and cultural influences, to better understand the nuanced ways in which gender shapes resilience [44].

The results of this study show a statistically significant difference between participants' social safeness and pleasure scores across different age groups (Table 4) (p<.05). Specifically, participants aged 25-27 scored significantly higher on the Social Safeness and Pleasure Scale compared to those in the 18-21 and 22-24 age groups. This finding suggests that social safeness and pleasure, which are integral components of psychological well-being, may improve as individuals mature. This age-related difference could be explained by several factors related to developmental psychology. Individuals in the 25-27 age group are more likely to have completed key stages of psychological and emotional development, as they transition from emerging adulthood into full adulthood [45]. This period often marks increased stability in personal identity, career goals, and social relationships [46]. As they advance in these areas, individuals may experience higher levels of self-confidence, security, and satisfaction in their social environments, leading to higher levels of social safeness and pleasure. In contrast, individuals in the younger age groups (18-21 and 22-24) may still be navigating these developmental milestones. Adolescents and young adults are often in a phase of exploration, facing challenges related to identity formation, career choices, and establishing long-term relationships [47]. This transitional period may be associated with greater stress and lower levels of perceived social safety and pleasure, as they have not yet achieved the same level of personal and social maturity as older adults. Supporting this perspective, research by Reed et al. on resilience across different age groups also highlights the differences in psychological outcomes depending on age [36]. In their study, men in the 20-29 age group had significantly higher resilience scores compared to women, suggesting that young adults may experience resilience and social safeness differently depending on their life stage. However, no significant gender differences were found in other age groups, which supports the idea that age-related developmental factors can significantly influence psychological outcomes [48]. Moreover, the findings of this study align with the broader literature on age-related changes in psychological well-being. As people age and mature, they often gain better coping mechanisms, social support systems, and a sense of personal achievement, all of which contribute to improved social safeness and pleasure [49]. These elements are critical for fostering resilience and well-being in adulthood, particularly in comparison to the challenges often faced by younger individuals in their early stages of life.

It is observed that there is a statistically significant difference in terms of the total dimensions of the participants' resilience and the variable of the faculties studied (Table 5) (p<.05). A statistically significant difference was detected in favour of the Faculty of Sports Sciences in terms of the total dimensions of resilience according to the faculties variable. This situation reveals that the participants studying at the Faculty of Health Sciences and the Faculty of Islamic Sciences have a lower level of resilience than the participants studying at the Faculty of Sports Sciences. In addition, it is seen that there is a statistically significant difference (Table 4) in terms of the faculties variable in the Emotional Coping and Situational Coping dimensions, which are among the sub-dimensions of resilience. Considering the reason for this, it can be said that the faculties of sports sciences, health sciences and Islamic sciences have different perspectives on students physically and mentally. While sports science students strengthen their bodies physiologically through physical activity and exercise, health sciences students gain knowledge and skills to protect and improve people's health by specialising in health issues.

Students of the faculty of Islamic sciences strive to understand and live the teachings of the Islamic religion. When the literature findings were examined, it was determined that there were studies supporting the results of our study and studies stating the contrary. For example, Sezgin found in his research that there was no significant difference in students' psychological resilience levels depending on the faculty they studied at [18]. Likewise, in the study by Eryıl-maz examining the psychological resilience of university students, he did not detect a signifi-cant difference in the psychological resilience levels of the students according to the faculty variable [50]. Atarbay concluded in his study that the undergraduate programs in which uni-versity students study do not significantly affect their psychological resilience [51]. Atan et al., in their study conducted by Ondokuz Mayıs University, in the statistical comparison of the psychological resilience scale sub-dimension and total scores of students from the Faculty of Sports Sciences and the Faculty of Theology, according to gender, female students' future per-ception, family harmony, social competence, social [52]. It was determined that the resources sub-dimension and total scores were significantly higher than the scores of male students.

The results of this study indicate a statistically significant difference in the social safeness and pleasure levels of participants across different departments (Table 6) (p<.05). Specifically, students in the Coaching Department reported higher levels of social safeness and pleasure compared to students in the Nursing and Sports Management departments. This difference may be attributed to several factors related to the specific nature of each department and its curriculum, as well as the students' active engagement in physical activities. The Coaching Department students are likely to experience higher levels of social safeness and pleasure due to the strong emphasis on physical activity and sports participation embedded within their academic training. Research has shown that regular engagement in physical exercise and sports activities contributes significantly to psychological well-being, increasing both physical and mental health [53]. These students' involvement in active sports may not only foster physi-cal health but also promote social interaction, community engagement, and personal satisfac-tion—elements that are vital components of social safeness and pleasure [54]. Furthermore, the Coaching curriculum often focuses on teamwork, leadership, and communication, which could enhance feelings of social connectedness and emotional well-being. In contrast, students in the Nursing and Sports Management departments, while also engaging with their respective fields, may not have the same level of active sports involvement. Nursing students, for example, focus more on healthcare practices and caregiving skills, which, though deeply rewarding, may not offer the same degree of physical and social interaction as sports-based education. Simi-larly, Sports Management students are often involved in the organizational and administrative aspects of sports, which may not directly involve them in the physical activity and social inter-actions that are key to fostering social safeness and pleasure. As a result, their levels of social safeness and pleasure may be comparatively lower, which aligns with the findings in this study. Moreover, this result is consistent with previous research exploring the relationship between social safeness, pleasure, and departmental influences. For instance, studies have found that the academic environment and curriculum can influence students' social connectedness and life satisfaction [55]. In a study conducted in China, it was shown that people's sense of social trust and overall life satisfaction were significantly influenced by the quality of their educa-tional or institutional experiences, including their perceptions of fairness and support within their community [56]. This suggests that the social context created within different academic departments can have a substantial impact on students' psychological well-being, particularly regarding feelings of safety and pleasure. Furthermore, the higher levels of social safeness and pleasure reported by Coaching students could also be linked to the close-knit community often fostered in sports-related departments. As noted by Putnam, participation in communal activities, such as team sports or physical fitness training, can lead to stronger social networks

and higher levels of trust, both of which are essential for a sense of social safety [57]. The dynamics within the Coaching Department, therefore, may contribute positively to these students' social and emotional development. It is seen that there is a statistically significant difference in the social safeness and pleasure levels of the participants in the study in terms of the department variable. This difference shows that the Coaching Department is superior to Nursing and Sports Management. If we look at the reason for this, it can be said that the level of active sports participation of students in the Coaching department may be due to the content of their courses. Wang et al. examined the quality of social security systems and found that satisfactory social security significantly increases institutional and interpersonal trust [11]. Their research highlights the role of governance quality in influencing satisfaction levels across different social strata, including various professional and social departments. Zhao et al. found that social trust and life satisfaction positively correlate with perceptions of social security fairness [58]. Their research indicates that departments or groups with differing levels of trust in social security systems may exhibit varying levels of overall life satisfaction. It is seen that there is a statistically significant difference in terms of the department variable of the resilience of the participants in the study. This difference appears to be superior to that of the Physical Education and Sports Teaching Department compared to that of the Nursing Department. In addition, it is seen that the difference in the Coaching department is superior to the nursing and sports management departments. Let's look at the reason for this superiority from the perspective of the Physical Education and Sports Teaching Department. It can be said that the students studying in this department are more interested in sports and the pedagogical training they receive while studying also has an effect. We can make the same comment that it affects why students studying in the Coaching department are superior to those studying in the nursing and sports management departments. Nguyen et al. explored resilience among L2 learners and found significant differences in resilience levels across various academic departments [59]. The research indicated that students in departments with higher stress and more demanding curricula showed greater resilience due to frequent exposure to academic challenges than those in less demanding fields. Martin et al. investigated resilience and academic buoyancy among Australian high school learners[60]; this study noted significant differences between students in different academic streams. Those in higher-pressure academic tracks exhibited higher resilience scores due to continuous exposure to academic stressors and the necessity for adaptive coping strategies.

The current study reveals a statistically significant difference in resilience and sports status variables among the participants (Table 7) ($p < .05$). Specifically, the findings suggest that individuals who engage in regular sports or physical activity demonstrate significantly higher resilience levels compared to those who do not participate in physical activity. This result aligns with a growing body of research suggesting that physical activity plays a pivotal role in enhancing psychological resilience. Physical activity is known to exert positive effects on various psychological outcomes, particularly resilience. One key reason for this is that physical activity involves pushing the body's limits, which can translate into psychological benefits. As individuals engage in regular sports or exercise, they often face and overcome physical challenges, fostering a sense of accomplishment and self-efficacy. These experiences may increase one's ability to manage stress and navigate adversity in everyday life. According to a study by Zlatar et al., physical activity not only boosts physical well-being but also enhances resilience by reinforcing psychological flexibility and emotional regulation [61]. This can be particularly significant in the context of stress management, as individuals accustomed to pushing their physical limits tend to develop greater psychological endurance when faced with life challenges. Additionally, a study conducted on college students found a positive correlation between physical activity and resilience, with the satisfaction of basic psychological needs—competence,

autonomy, and relatedness—acting as a mediator in this relationship. These basic needs, which are central to Self-Determination Theory [62] are often fulfilled through regular participation in physical activities, contributing to an enhanced sense of competence and autonomy, which in turn strengthens resilience. When individuals feel more capable in their physical and psychological abilities, they are more likely to handle stress and setbacks effectively. Furthermore, the authors found that individuals who maintained physical activity routines during the pandemic showed better psychological outcomes, including greater resilience in coping with the stressors imposed by lockdowns and isolation. This supports the notion that physical activity can serve as a protective factor against stress and may help individuals maintain psychological well-being during challenging times. However, it is important to note that not all studies have consistently supported these findings. Some research has found no significant differences in resilience based on physical activity levels. For example, a study by Wagnild and Young examining resilience across various demographic groups found no significant variations in resilience scores between individuals categorized as active and inactive when other variables, such as age and gender, were controlled for [63]. This suggests that while physical activity may contribute to resilience, other factors—such as social support, personality traits, and life experiences—can also play critical roles in shaping an individual's resilience levels. In summary, while the current study supports the positive link between physical activity and resilience, these findings should be considered in the context of the broader literature, which indicates that resilience is influenced by a variety of factors. The relationship between sports participation and resilience may be more complex than a direct cause-effect, with psychological factors such as motivation, emotional regulation, and social support playing significant moderating roles. This finding suggests that individuals who engage in physical activity may possess higher levels of psychological resilience, which enables them to better cope with challenges encountered in daily life. One possible explanation is that participating in sports involves pushing physiological limits, which could help build mental toughness and emotional endurance. As individuals confront and overcome physical challenges, they may develop greater self-confidence, emotional regulation, and stress management skills, all of which contribute to resilience. Previous studies have supported this notion. For instance, a study examining the effect of physical activity on resilience among university students found a positive and significant association between physical activity and resilience. Specifically, the study demonstrated that engaging in physical activities can enhance resilience by satisfying basic psychological needs such as autonomy, competence, and relatedness. These findings are consistent with the principles of Self-Determination Theory, which posits that when individuals meet these psychological needs, they experience greater well-being and are better equipped to deal with adversity [64]. In particular, sports participation can foster a sense of competence and autonomy, which, in turn, may enhance an individual's capacity to recover from stress and setbacks. Furthermore, the study's results align with other research that suggests physical activity plays a vital role in strengthening resilience. For example, a study by Reed et al. revealed that physical activity significantly improved participants' resilience levels, with active individuals demonstrating higher resilience scores compared to their sedentary counterparts [36]. This study highlights the potential psychological benefits of sports, suggesting that regular engagement in physical activities can serve as a protective factor against stress and help build psychological resilience over time. Overall, the current study's findings, along with previous research, emphasize the importance of physical activity as a key contributor to resilience. By promoting sports participation, institutions and educators can potentially help individuals develop the psychological skills needed to cope with stress, adapt to challenges, and maintain emotional well-being in various aspects of life.

The results from the Pearson Product Moment Correlation Analysis (Table 8) indicate a statistically significant positive relationship between the scores from the Social Safeness and Pleasure Scale and the total and sub-dimension scores of the Resilience Scale, with significance at the $p < .01$ level. Specifically, the analysis reveals that higher scores in the Social Safeness and Pleasure scale correlate positively with increased scores in the sub-dimensions of resilience, including social support, emotional coping, and situational coping, all at a $p < .01$ significance level. This finding aligns with previous studies that suggest a close link between social and emotional well-being and resilience. For instance, research by Luthar et al. emphasized that resilience is a multifaceted construct, deeply rooted in an individual's ability to access and utilize social support networks, particularly in coping with challenging situations [65]. The significant positive relationship between social safeness, pleasure, and the social support dimension of resilience further corroborates these findings, suggesting that individuals who feel secure in their social environment are better equipped to seek and utilize support during times of stress. Moreover, the connection between social safeness and emotional coping as well as situational coping aligns with studies examining the role of positive emotions and social environments in building resilience. For example, Tugade and Fredrickson demonstrated that positive emotions, often fostered through secure social connections, play a critical role in enhancing individuals' ability to cope with adversity [66]. Social safeness, which encapsulates a sense of security and emotional well-being, thus contributes to an individual's capacity for emotional regulation and adaptive coping strategies in diverse situations. The results also support the notion that positive emotional states, as facilitated by social safeness, may empower individuals to effectively navigate stressful situations, as suggested by Fredrickson's broaden-and-build theory of positive emotions [67]. According to this theory, positive emotions broaden cognitive and behavioral repertoires, which, in turn, build resilience by enhancing individuals' coping skills and fostering a more resourceful response to life's challenges. In summary, the significant correlations found in this study highlight the vital role of social safeness and pleasure in fostering resilience, particularly in terms of social support, emotional coping, and situational coping. These findings contribute to the growing body of literature emphasizing the interdependence of emotional well-being, social connections, and resilience. Future research could further explore the underlying mechanisms by which social safeness influences resilience, particularly in different populations or under varying levels of stress.

## Conclusion

This study aimed to determine students' social safeness, pleasure, and resilience levels in the faculties of Sports Sciences, Health Sciences and Islamic Sciences. It was observed that the participants generally had social safeness and a level of pleasure above the mid-level. No significant differences were detected in variables such as gender, study faculty, and sports

**Table 8.  Table of results of pearson product moment correlation analysis conducted to determine the relationship between the scores obtained from the social safeness and pleasure scale and the scores of the resilience scale.**

| Social Safeness And Pleasure | N | R | P |
|---|---|---|---|
| Resilience Total | 350 | ,652** | ,001 |
| Social Support | 350 | ,319** | ,001 |
| Emotional Coping | 350 | ,512** | ,001 |
| Situational Coping | 350 | ,547** | ,001 |

**. Correlation is significant at the 0.01 level (1-tailed).

participation. However, significant differences were observed regarding age, department and opinion that working with a sports psychologist was beneficial. When the resilience Strength levels were examined, it was found that the participants generally had a level above the mid-level. No significant differences were observed in the case of thinking that working with a sports psychologist was beneficial. Still, significant differences were determined in some variables such as gender, age, faculty, department and sports activity. In summary, this study determined students' social safety, pleasure, and self-recovery levels. Students were generally found to be above intermediate level in these areas. However, it has been suggested that some variables differ and should be examined in more detail.

## Recommendations for future research

This study has provided insights into the relationships between social safeness and pleasure and resilience levels among university athletes. However, to deepen the understanding of these relationships, further research is encouraged in the following areas:

1. Future research could explore how social safeness, pleasure, and resilience manifest in athletes from different types of sports (e.g., team vs. individual sports) and competitive levels (e.g., amateur vs. professional). Such comparative studies would help clarify the influence of varying sports dynamics on these psychological constructs.

2. Given that resilience and social safeness can fluctuate over time, longitudinal studies are recommended to capture temporal changes and causative patterns within these variables. Experimental designs incorporating targeted interventions to enhance social safeness and pleasure, such as social skills training and positive psychology interventions, may also provide valuable insights into effective strategies for resilience development in athletes.

3. Investigating these relationships across different cultural backgrounds could enhance the generalizability of findings and offer a more comprehensive understanding of the role cultural context plays in the development of resilience, social safeness, and pleasure among athletes.

4. Psychophysiological Measures: To objectively measure resilience and stress responses, psychophysiological assessments, such as heart rate variability, could be integrated. These measures would provide valuable physiological insights that complement self-reported data, contributing to a more holistic evaluation of athletes' psychological well-being.

By addressing these recommendations, future studies could advance the field of sport psychology and enhance the support systems for athletes, promoting both mental resilience and overall well-being.

## Supporting information

**S1 File. Contact ınformation for data access committee.** Details for accessing data through the Data Access Committee.
(DOCX)

**S1 Fig. Dataset.** The raw dataset used in the study (in Excel format). Data has been anonymized to comply with confidentiality principles.
(XLSX)

**S2 Fig. Dataset.** The raw dataset used in the study (in SPSS format). Data has been anonymized to comply with confidentiality principles.
(SAV)

## Acknowledgments

The authors would like to thank the institution officials who contributed to the regular maintenance of data collection and implementation. Finally, we would like to thank all our youth for their contributions.

## Author Contributions

**Conceptualization:** Veysel Temel, Nurhan Hümeyra Özçelik.

**Data curation:** Veysel Temel, Nurhan Hümeyra Özçelik.

**Formal analysis:** Veysel Temel, Nurhan Hümeyra Özçelik.

**Investigation:** Veysel Temel, Nurhan Hümeyra Özçelik.

**Methodology:** Veysel Temel, Nurhan Hümeyra Özçelik.

**Resources:** Veysel Temel, Nurhan Hümeyra Özçelik.

**Supervision:** Veysel Temel, Nurhan Hümeyra Özçelik.

**Validation:** Nurhan Hümeyra Özçelik.

**Writing – original draft:** Veysel Temel, Nurhan Hümeyra Özçelik.

**Writing – review & editing:** Veysel Temel, Nurhan Hümeyra Özçelik.

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

       066X.56.3.218

