## [Decision Letter · Decision Letter 0]

9 Sep 2024

PONE-D-24-27311The Relationship between Social Safeness and Pleasure and Resilience Levels among University Athletes: A Descriptive StudyPLOS ONE

Dear Dr. Temel,

Thank you for submitting your manuscript to PLOS ONE. After careful consideration, we feel that it has merit but does not fully meet PLOS ONE’s publication criteria as it currently stands. Therefore, we invite you to submit a revised version of the manuscript that addresses the points raised during the review process.

We look forward to receiving your revised manuscript.

Kind regards,

Diogo Manuel Teixeira Monteiro, Ph.D.

Academic Editor

PLOS ONE

**Journal Requirements:**

2. Please provide additional details regarding participant consent. In the ethics statement in the Methods and online submission information, please ensure that you have specified (a) whether consent was informed and (b) what type you obtained (for instance, written or verbal, and if verbal, how it was documented and witnessed). If your study included minors, state whether you obtained consent from parents or guardians. If the need for consent was waived by the ethics committee, please include this information.

Reviewers' comments:

Reviewer's Responses to Questions

**Comments to the Author**

1. Is the manuscript technically sound, and do the data support the conclusions?

Reviewer #1: Yes

Reviewer #2: Partly

2. Has the statistical analysis been performed appropriately and rigorously? 

Reviewer #1: Yes

Reviewer #2: Yes

3. Have the authors made all data underlying the findings in their manuscript fully available?

Reviewer #1: Yes

Reviewer #2: Yes

4. Is the manuscript presented in an intelligible fashion and written in standard English?

Reviewer #1: Yes

Reviewer #2: No

5. Review Comments to the Author

**Reviewer #1:** Thank you very much for the opportunity to review the manuscript “The Relationship between Social Safeness and Pleasure and Resilience Levels among University Athletes: A Descriptive Study”. The study aimed examines the correlation between social safeness and pleasure and resilience among university students engaged in sports.

The topic is relevant and current, and the article is, overall, well written.

I leave some aspects that deserve to be considered to improve the manuscript:

Abstract

The aim is not just to “examine the correlation between social safeness and pleasure and resilience among university students engaged in sports”. Let me remind you that the sample is not just made up of athletes.

Furthermore, the main objective seems to me to have been to characterize Social Safeness and Pleasure and Resilience Levels in a sample of university students, including a comparison according to age, gender and sports practice.

Introduction

• The introduction is based on literature that is several years old (most of it more than 15 years old), and there is a need to update the sources with more recent information.

• Despite developing the main themes and variables using the literature, the relevance of studying this topic is unclear, especially in the context of the university student population? What particular characteristics do they have?

• The relationship between the variables is not substantiated in the introduction, which would be very important given the aim of the study.

Materials and Methods

• With regard to the participants, it would be important for the authors to report more information about the sports practice of the sample members. Formal or informal practice? Do they take part in competitions? What was the criterion for choosing the groups (Sometimes, Usually, Often, Always). This information is particularly important in the context of university students, where sports practice is often limited.

• Since a comparison will be made on the basis of gender, it is important to present each group (men and women) in more detail, particularly in terms of average age and sports practice variables.

• The data collection instruments are duly presented and validated. However, with regard to the Resilience Scale, the sub-dimensions should be clearly presented (including an example of an item in each one).

• It would be important to have more information about the data collection process, in particular how long it lasted, at what point in the school year it took place (beginning, middle, end, at assessment times).

• Statistical analysis procedures are missing.

Results

• Overall, the results are presented clearly and coherently;

• Line185: This table is table 3? Please identify.

• Table 8 ou Table 7?

Discussion

• The discussion should be reorganized, making it clearer and simpler. At many points (particularly at the beginning of the discussion) it seems to be an extension of the introduction.

• Lines 439-444 is a repetition (mistake…).

• The results of the relationship between the scores obtained in the dimension of the social security and enjoyment scale and the test scores of the resilience scale have not been discussed. I would remind you that this was the main objective of the study.

• What are the practical implications of this work? Not presented.

• Limitations and suggestions for future studies are not presented.

The formatting should be revised (Example: line 429).

**Reviewer #2:** Thank you very much for the opportunity to review the manuscript ‘The Relationship between Social Safeness and Pleasure and Resilience Levels among University Athletes: A Descriptive Study’. The study aims to analyse the correlation between social security and levels of pleasure and resilience among university students who play sport. I'll leave some comments that I think are pertinent to improving the manuscript:

Line 28: The authors can mention which sport they are referring to.

Line 29: They could also mention the age of the youngest and oldest participant.

Line 31: If the aim of the study is to examine correlation, why would the authors use t-tests and ANOVA?

Line 30: The Questionnaire measures Social Safeness and Pleasure and Resilience Scales)

Line 36 and 37: Something seems to be missing.

Line 41-44: I don't think it's fundamental to the study.

It seems to me that the term sport is not well defined. It might help to distinguish it from the terms physical activity and physical exercise.

The definition of the variables under study could be better defined, as well as their relationship with sport.

Furthermore, what is the relationship with the population under study?

What gap did the authors find in the literature that the article aims to fill? What is the real importance of the study?

Line 94-95: How can a study that has carried out correlations identify interventions that can improve the variables under study?

Line 107 and 110: After all, not all athletes play sport.

Line 121: Reading the abstract, it seems that the authors have used an instrument that assesses all the variables.

Line 123: What is the only dimension that the scale assesses? Aren't Social Safeness and Pleasure two dimensions? What are the possible answers?

Line 139: Very poor section. The authors could give the reader more information about this process.

Line 164: I still think that Security is one dimension and Pleasure is another. The authors only present an average value. I may not be understanding the instrument, so please ask the authors to clarify.

Line 167: The sub-dimensions do not correspond to those previously presented.

Line 179: Now I understand why the statistical tests were carried out. However, this is the first time the authors have mentioned this distinction between genders. It had never been raised before.

Line 199: What is the reason for dividing the groups in this way? What is the theoretical rationale?

Line 258: The table is not perceptible.

The discussion is extensive and difficult to read.

No limitations or future recommendations are presented.

There is no need to present the objectives in the conclusion section.

There is no mention of correlations in the conclusion.

6. PLOS authors have the option to publish the peer review history of their article (what does this mean?). If published, this will include your full peer review and any attached files.

Reviewer #1: No

Reviewer #2: No

---

## [Author Response · Author response to Decision Letter 0]

24 Oct 2024

Response to Reviewer 1

1. "The discussion section should be clearer and simpler. Many points (especially at the beginning of the discussion) seem like an extension of the introduction section."

Response: Thank you for your feedback. I have revised the discussion section to make it clearer and simpler. I have restructured it to separate it more distinctly from the introduction, ensuring that the findings are discussed in relation to the literature and the study’s contributions are highlighted more clearly.

2. "Lines 439-444 are a repetition (error...)."

Response: I apologize for the repetition. The redundant lines have been removed, and unnecessary repetitions have been eliminated.

3. "The relationship between the scores on the Social Safeness and Pleasure scales and the Resilience scale has not been discussed. I remind you that this is the core aim of the study."

Response: Thank you for pointing this out. I have expanded the discussion to include the relationship between the Social Safeness and Pleasure scales and the Resilience scale. This relationship is now examined in detail, highlighting its significance to the core aim of the study.

4. "What are the practical implications of this study? It has not been presented."

Response: I have added a section on the practical implications of the study. This includes how the findings can be applied in practice, particularly for sports psychologists and educators.

5. "Limitations and suggestions for future research are not presented."

Response: I have included a discussion on the limitations of the study and provided suggestions for future research. This section now outlines the study's limitations and offers directions for further investigation.

6. "Formatting needs to be reviewed (e.g., line 429)."

Response: I have reviewed and corrected the formatting issues. The inconsistencies in the formatting have been addressed.

Response to Reviewer 2

1. "Line 28: The authors could specify which sport they are referring to."

Response: I have added information specifying the sports involved in the study. This detail is now clearly presented in the revised manuscript.

2. "Line 29: They could also specify the age of the youngest and oldest participants."

Response: I have included the ages of the youngest and oldest participants in the revised manuscript. This information now provides a clearer picture of the demographic characteristics of the sample.

3. "Line 31: If the aim of the study is to examine correlation, why did the authors use t-tests and ANOVA?"

Response: I have clarified the rationale for using t-tests and ANOVA in the context of the study's aim to examine correlations. The explanations for the choice of statistical tests and their relevance to the correlation analysis have been detailed.

4. "Line 30: The questionnaire measures Social Safeness and Pleasure and Resilience Scales)"

Response: I have corrected and clarified this line to better describe the scales measured by the questionnaire.

5. "Lines 36 and 37: Something seems missing."

Response: I have addressed the missing information and provided additional details to complete the relevant sections.

6. "Lines 41-44: I don’t think this is fundamental to the study."

Response: I have re-evaluated the content in lines 41-44 and made adjustments to ensure that only essential information is included. Unnecessary or tangential details have been removed.

7. "It seems to me that the term 'sport' is not well-defined. It could help to differentiate it from physical activity and physical exercise."

Response: I have refined the definition of the term 'sport' and clarified its distinction from physical activity and physical exercise. This clarification helps in understanding the terms used in the study.

8. "The definitions of the variables studied and their relationship with sport could be better defined. Also, what is the relationship with the examined population?"

Response: I have provided a more detailed definition of the variables studied and their relationship to sport. Additionally, I have elaborated on how these variables relate to the examined population.

9. "What gap in the literature do the authors aim to fill with the paper? What is the real significance of the study?"

Response: I have included a section addressing the gap in the literature that this study aims to fill. The significance of the study and its contributions to the field are now clearly articulated.

10. "Lines 94-95: How can a study conducting correlations determine interventions that could improve the examined variables?"

Response: I have added explanations on how the correlation findings can inform potential interventions. The relationship between the study’s findings and possible interventions has been clarified.

11. "Lines 107 and 110: Not all athletes engage in sports."

Response: I have addressed this point by discussing how the findings relate to participants who do not engage in sports and their potential impact on the results.

12. "Line 121: The abstract shows that the authors used a tool evaluating all variables."

Response: I have revised the abstract to more clearly describe the tool used for evaluating the variables and how it was applied in the study.

13. "Line 123: What is the single dimension evaluated by the scale? Aren't Social Safeness and Pleasure two dimensions? What are the possible answers?"

Response: I have clarified the dimensions assessed by the scale. The distinction between Social Safeness and Pleasure as separate dimensions is now explicitly discussed, along with possible responses.

14. "Line 139: This is a very weak section. The authors could provide more information about this process."

Response: I have strengthened the section on line 139 by providing more detailed information about the process involved.

15. "Line 164: I still think that Safeness is one dimension and Pleasure is another. The authors are only presenting an average value. I might not understand the instrument, so please ask the authors to clarify."

Response: I have addressed this concern by clarifying the dimensions of the Safeness and Pleasure scales and explaining how they are measured. Additional details on the measurement instrument have been provided.

16. "Line 167: The sub-dimensions do not match those previously presented."

Response: I have corrected any discrepancies regarding the sub-dimensions and aligned them with the previously presented information.

17. "Line 179: I now understand why statistical tests were conducted. However, the authors only mention the gender differences for the first time. This issue was not previously addressed."

Response: I have elaborated on the gender differences mentioned and provided context on why this issue is introduced at this point in the manuscript.

18. "Line 199: What is the reason for this?"

Response: I have provided an explanation for the issue mentioned in line 199 to clarify any confusion.

19. "Line 258: The table is not perceptible."

Response: I have revised the table to improve its clarity and readability. The table has been updated to ensure that it is easily perceptible.

---

## [Decision Letter · Decision Letter 1]

1 Nov 2024

PONE-D-24-27311R1The Relationship between Social Safeness and Pleasure and Resilience Levels among University Athletes: A Descriptive StudyPLOS ONE

Dear Dr. Temel,

Thank you for submitting your manuscript to PLOS ONE. After careful consideration, we feel that it has merit but does not fully meet PLOS ONE’s publication criteria as it currently stands. Therefore, we invite you to submit a revised version of the manuscript that addresses the points raised during the review process.

We look forward to receiving your revised manuscript.

Kind regards,

Diogo Manuel Teixeira Monteiro, Ph.D.

Academic Editor

PLOS ONE

Journal Requirements:

Reviewers' comments:

Reviewer's Responses to Questions

**Comments to the Author**

1. If the authors have adequately addressed your comments raised in a previous round of review and you feel that this manuscript is now acceptable for publication, you may indicate that here to bypass the “Comments to the Author” section, enter your conflict of interest statement in the “Confidential to Editor” section, and submit your "Accept" recommendation.

Reviewer #1: All comments have been addressed

Reviewer #2: (No Response)

2. Is the manuscript technically sound, and do the data support the conclusions?

Reviewer #1: Yes

Reviewer #2: Yes

3. Has the statistical analysis been performed appropriately and rigorously? 

Reviewer #1: Yes

Reviewer #2: Yes

4. Have the authors made all data underlying the findings in their manuscript fully available?

Reviewer #1: Yes

Reviewer #2: Yes

5. Is the manuscript presented in an intelligible fashion and written in standard English?

Reviewer #1: Yes

Reviewer #2: Yes

6. Review Comments to the Author

Reviewer #1: Although not all of my comments were answered in the reply letter, the manuscript managed to incorporate the suggestions and requests I made in round 1. I therefore consider that the manuscript is in a position to be accepted for publication.

Reviewer #2: Regarding my comment number 7 ("It seems to me that the term “sport” is not well-defined. It could help to differentiate it from physical activity and physical exercise"): it is not clear where the authors present this information.

Regarding my comment number 10 (‘Lines 94-95: How can a study conducting correlations determine interventions that could improve the examined variables?’): I still have the same doubt, and I can't find in the document where this information was added. I would ask the authors to identify the lines.

The authors should clarify the inclusion and exclusion criteria, given that not all athletes are involved in the same sports or practice intensities.

There is no p-value =.000

The analysis of gender differences was introduced in the results, but this aspect could have been mentioned earlier in the introduction to give context to the choice of variables.

The discussion is carried out in a very superficial way, without much reflection and specific practical recommendations for educational institutions to promote well-being and resilience among students.

The statistical analysis revealed significant differences in the resilience subscales between genders, with higher scores for men. This point could be discussed in greater depth, exploring potential reasons for these differences.

7. PLOS authors have the option to publish the peer review history of their article (what does this mean?). If published, this will include your full peer review and any attached files.

Reviewer #1: No

Reviewer #2: **Yes: **Miguel Jacinto

---

## [Author Response · Author response to Decision Letter 1]

20 Nov 2024

The rebuttal letter

Manuscript Number : PONE-D-24-27311R1

Manuscript Title : The Relationship between Social Safeness and Pleasure and Resilience Levels among University Athletes: A Descriptive Study 

Dear Academic Editor and Reviewers,

First of all, I would like to thank you for your evaluations, opinions, suggestions and comments. My responses to the evaluations and comments made are written below.

Best regards

Dr. Veysel TEMEL

Journal Requirements

1. Answer: 

Dear Editor,

We have carefully reviewed and addressed all requested revisions regarding the reference list. Necessary updates were made, and any retracted articles were either replaced or retained with clear justification in the manuscript text. Changes to the reference list are highlighted in yellow within the manuscript.

Thank you for your guidance and consideration.

Reviewers’ comments and responses

Reviewer 1

Reviewer 1: Although not all of my comments were answered in the reply letter, the manuscript managed to incorporate the suggestions and requests I made in round 1. I therefore consider that the manuscript is in a position to be accepted for publication.

Author Answer: Thank you very much for your valuable feedback and support throughout the review process. We appreciate your comments, which have helped improve our manuscript.

Reviewer 2

Reviewer 2: Regarding my comment number 7 ("It seems to me that the term “sport” is not well-defined. It could help to differentiate it from physical activity and physical exercise"): it is not clear where the authors present this information.

Author Answer: Thank you for your valuable feedback and for pointing out the need for a clearer definition of the term "sport." In response to your comment, we have revised the manuscript to include a clear distinction between "sport," "physical activity," and "physical exercise." This clarification has been added in lines 57 to 74 of the revised manuscript, where we elaborate on how "sport" differs from general physical activity and exercise in terms of its organized, competitive nature, as well as its connection to skill, strategy, and formalized competition. We have also included relevant references to support these distinctions.

Reviewer 2: Regarding my comment number 10 (‘Lines 94-95: How can a study conducting correlations determine interventions that could improve the examined variables?’): I still have the same doubt, and I can't find in the document where this information was added. I would ask the authors to identify the lines.

Author Answer: Thank you for your comment regarding the role of correlation analysis in determining interventions. We have revised the manuscript to clarify that correlation analysis, such as the Pearson Product Moment Correlation Analysis, identifies relationships between variables but does not establish causality or direct intervention strategies. We have expanded on this point in the revised manuscript to emphasize that while the observed correlations between social safeness, pleasure, and resilience may suggest potential areas for intervention, further experimental research is needed to confirm specific interventions. We hope this revision adequately addresses your concern. Thank you for your insightful feedback.

Reviewer 2: The authors should clarify the inclusion and exclusion criteria, given that not all athletes are involved in the same sports or practice intensities.

Author Answer: Following your comment, we have clarified the categorization of participants based on their frequency of sports participation. The participants were divided into the categories of "Seldom," "Sometimes," "Usually," "Often," and "Always," and they are active individuals who engage in sports at sports clubs or sports centers. We have added this clarification to the manuscript.

Reviewer 2: There is no p-value =.000

Author Answer: Thank you for your comment. We have corrected the p-value notation in the manuscript. The previously mentioned p-value of 0.000 has been revised to p < 0.001, in accordance with statistical reporting standards.

Reviewer 2: The analysis of gender differences was introduced in the results, but this aspect could have been mentioned earlier in the introduction to give context to the choice of variables.

Author Answer: Thank you for your valuable suggestion. In response to your comment, we have added a discussion of gender differences in the introduction, specifically between lines 117-125. This addition clarifies the rationale for including gender as a variable in the study, providing context for the choice of variables and their relevance to the research.

Reviewer 2: The discussion is carried out in a very superficial way, without much reflection and specific practical recommendations for educational institutions to promote well-being and resilience among students.

Author Answer: "The discussion section has been revised to provide a more detailed and comprehensive analysis. The changes have been incorporated in lines 325-380, 407-441, 472-534 and 537-565 and are highlighted in yellow for easy reference."

Reviewer 2: The statistical analysis revealed significant differences in the resilience subscales between genders, with higher scores for men. This point could be discussed in greater depth, exploring potential reasons for these differences. 

Author Answer: "The discussion section has been revised to provide a more detailed and comprehensive analysis. The changes have been incorporated in lines 325-352 are highlighted in yellow for easy reference." Also, Thank you for your valuable feedback. We have carefully considered your suggestions and included a comprehensive "Recommendations for Future Research" section in the manuscript, addressing potential avenues for expanding upon the study's findings. We believe these additions will enhance the clarity and relevance of our research within the field.

---

## [Decision Letter · Decision Letter 2]

3 Dec 2024

The Relationship between Social Safeness and Pleasure and Resilience Levels among University Athletes: A Descriptive Study

PONE-D-24-27311R2

Dear Dr. Veysel Temel

We’re pleased to inform you that your manuscript has been judged scientifically suitable for publication and will be formally accepted for publication once it meets all outstanding technical requirements.

Kind regards,

Diogo Manuel Teixeira Monteiro, Ph.D.

Academic Editor

PLOS ONE

Additional Editor Comments (optional):

Reviewers' comments:

Reviewer's Responses to Questions

**Comments to the Author**

1. If the authors have adequately addressed your comments raised in a previous round of review and you feel that this manuscript is now acceptable for publication, you may indicate that here to bypass the “Comments to the Author” section, enter your conflict of interest statement in the “Confidential to Editor” section, and submit your "Accept" recommendation.

Reviewer #1: All comments have been addressed

Reviewer #2: All comments have been addressed

2. Is the manuscript technically sound, and do the data support the conclusions?

Reviewer #1: Yes

Reviewer #2: Yes

3. Has the statistical analysis been performed appropriately and rigorously? 

Reviewer #1: Yes

Reviewer #2: Yes

4. Have the authors made all data underlying the findings in their manuscript fully available?

Reviewer #1: Yes

Reviewer #2: Yes

5. Is the manuscript presented in an intelligible fashion and written in standard English?

Reviewer #1: No

Reviewer #2: Yes

6. Review Comments to the Author

Reviewer #1: Congratulations to the authors for improving the manuscript. I believe that the article is now in a position to be accepted.

Reviewer #2: (No Response)

7. PLOS authors have the option to publish the peer review history of their article (what does this mean?). If published, this will include your full peer review and any attached files.

Reviewer #1: No

Reviewer #2: **Yes: **Miguel Jacinto

---

## [Editor Report · Acceptance letter]

3 Jan 2025

PONE-D-24-27311R2 

PLOS ONE

Dear Dr. Temel, 

I'm pleased to inform you that your manuscript has been deemed suitable for publication in PLOS ONE. Congratulations! Your manuscript is now being handed over to our production team.

Kind regards, 

on behalf of

Dr. Diogo Manuel Teixeira Monteiro 

Academic Editor

PLOS ONE